# ADAPTIVE INFERENCE: THEORETICAL LIMITS AND OPPORTUNITIES FOR EFFICIENT AI

## ABSTRACT

With the commercial deployment of increasingly larger and more complex neural networks at the cloud and the edge in recent years, inference has become too costly in terms of compute workload worldwide. Adaptive inference methods, which dynamically adjust a neural network's size or structure during inference, offer a means to enhance efficiency of neural networks beyond what static network compression and optimization methods can fundamentally achieve.

This paper introduces the first theoretical framework for quantifying the efficiency and performance gain opportunity size of adaptive inference algorithms. We provide new approximate and exact bounds for the achievable efficiency and performance gains, supported by empirical evidence demonstrating the potential for 10-100x efficiency improvements in both Computer Vision and Natural Language Processing tasks without incurring any performance penalties. Additionally, we offer insights on improving achievable efficiency gains through the optimal selection and design of adaptive inference state spaces.

## 1 INTRODUCTION

In recent years, neural networks have achieved human-level performance across various domains, ranging from image classification using vision transformers to intricate natural language processing tasks handled by Large Language Models. However, this notable improvement in performance comes with the caveat of training progressively larger models. The current high-performing vision transformers and large language models can only be effectively deployed on large cloud data-centers, leading to significant economical costs and environmental implications in terms of carbon footprint and energy consumption (Anthony et al., 2020; McDonald et al., 2022; Desislavov et al., 2023).

Currently, 80-90% of global cloud workloads consist of inference tasks (McDonald et al., 2022; Freund, 2019), and this percentage is expected to rise with the increased adoption of AI models. As models achieve peak performance and maturity, the demand for efficient inference has transitioned from a mere consideration to an immediate necessity (Samsi et al., 2023; Desislavov et al., 2023). This urgency is particularly heightened in non-cloud (edge) applications, where there is a demand for low latency execution of models on devices with limited memory, compute, and power resources (Xu et al., 2018; Li et al., 2019; Daghero et al., 2021).

The advent of network compression techniques, such as pruning and quantization, marked a significant stride in efficient inference. The initial achievements in this area paved the way for subsequent developments such as resource-aware neural architecture search, model distillation, and low-rank decomposition techniques, all aimed at enhancing the performance and efficiency of neural networks, either during training or as post-processing steps (Xu & McAuley, 2023; Li et al., 2023b; Han et al., 2023).

However, such efficiency advancements have reached a plateau, necessitating fundamentally new techniques that extend beyond the design space of conventional static neural network optimization methods.

One such technique is adaptive inference, founded on the intuition that for easier instances in the test-set, a simpler neural network might perform as accurately as a more complex one. Hence, an adaptive neural network (or an adaptive ensemble of networks), capable of dynamically adjusting its

complexity based on the difficulty of the input instance, can prove to be more efficient and, in some cases, even more accurate than an equivalent "static" model.

Adaptive inference methods, as commonly explored in the literature, make use of networks with dynamically tunable size and complexity (Han et al., 2021). Such networks are often adapted through techniques such as early exiting (Laskaridis et al., 2021; Ilhan et al., 2023; Yang et al., 2020) or through adaptive selection and mixture of experts (Meng et al., 2020; Li et al., 2023a; Jawahar & Mukherjee, 2023; Chen et al., 2023). One illustrative example is in context of Computer Vision (CV) is AR-Net (Meng et al., 2020), which showcases the use of a simple policy network during the inference phase to adaptively select between pre-trained classifiers with varying sizes and resolutions, achieving efficient video-based activity classification. Another example in context of Natural Language Processing (NLP) is the work by Rotem et al. (2023) comparing the performance and efficiency of both multi-model and early exiting approaches on large language models using BERT.

However, the adoption of adaptive inference methods for efficient AI has been limited compared to static network compression techniques. This can be mainly attributed to the ad-hoc nature of existing methods, and the lack of a comprehensive framework for designing adaptive inference data pipelines or gaining insight into the benefits as well as limitations of adaptive inference in specific tasks and applications.

This paper is the first to establish a theoretical foundation for analyzing adaptive inference methods and quantifying achievable efficiency and performance gains for general inference tasks. The proposed framework aims to bridge the gap between current ad-hoc methods and a more systematic approach to understanding and leveraging adaptive inference.

Our contributions encompass:

- A novel theoretical framework for the analysis of adaptive inference methods, achieved through the definition of conceptual Oracle Agents.

- Introduction of both approximate and exact bounds on the performance and efficiency gains achievable by an adaptive agent. This marks the inception of new quantitative measures for adaptation potential.

- Empirical findings showcasing adaptation potential and limits of models, demonstrated in the realms of both Computer Vision (Image Classification) and Natural Language Processing (Natural Language Inference).

- Design considerations and recommendations for maximizing efficiency and performance gain potential of neural networks.

- Ground truth "adaptation labels" for optimal adaptive inference, presented for two datasets and four neural networks in the context of image classification on ImageNet and Common-Sense NLI on HellaSwag.

## 2 A GENERAL FRAMEWORK FOR ADAPTIVE INFERENCE

As discussed earlier, adaptive inference is a broad term encompassing a variety of systems, applications, and methodologies. However, the majority of adaptive systems can be effectively abstracted as (finite) state machines (Hopcroft et al., 2001). A state machine is a conceptual framework that simplifies the behavior of dynamic systems by breaking down their complex dynamics into sets of "states" representing the system's behavior at specific points in time, and "transitions" depicting how the system evolves over time. This abstraction enables the separation of considerations and constraints imposed by the "adaptation state space" of a system from the performance of a specific "agent" responsible for guiding the system transitions between the states.

In this section, we begin by defining an adaptation state space within the context of a classification task. Subsequently, we present precise definitions and equations that explore the theoretical limits of performance and efficiency achievable by all possible adaptive inference agents. This is achieved through the analysis and definition of ideal "Oracle Agent"s.

## 2.1 MODEL ADAPTATION STATE SPACE FOR CLASSIFICATION

In classification-based adaptive inference, we often see the adaptation state space defined either using an ensemble of backbone classifiers (like in AR-Net (Meng et al., 2020), switching between different classifiers from EfficientNet family) or a single classifier with adaptable complexity (for instance, RA-Net (Yang et al., 2020) allowing different setups within a single backbone network).

For simplicity, we conceptualize both scenarios by representing them as a discrete set of $N$ backbone classifiers applied to a dataset $X$. These classifiers constitute a state space, defined as:

$$S = \{S_i\} \text{ for } i \in \{1, 2, 3, ..., N\}. \tag{1}$$

Suppose further that these classifiers are ranked based on the amount of resources they consume in an increasing order. In other words, let $R_i$ represent the resource consumption of the classifier in the $i$-th state $S_i$, then:

$$R_1 \leq R_2 \leq ... \leq R_i \leq ... \leq R_N. \tag{2}$$

In this definition, it is important to highlight that $R_i$ encompasses the total cost of selecting state $S_i$. This includes not only the classification compute cost but also potential resource consumption overhead of loading/reloading neural network weights or signal routing which can also be a function of the model size. (For an example of how to incorporate system-specific adaptation resource consumption overheads into the calculated bounds, refer to Section 4.2).

Let $A_i$ represent the test accuracy of classifier $i$ represented with $S_i$. For each state $S_i$, there exists a pair $(R_i, A_i)$, representing both the state's total resource consumption and the accuracy of the corresponding classifier. In practical systems, larger (and more resource-intensive) models are typically employed only if on average they deliver better or equal performance compared to smaller models. Consequently, we assume that the model accuracies follow an increasing order

$$A_1 \leq A_2 \leq ... \leq A_i \leq ... \leq A_N. \tag{3}$$

Given the definition of the adaptation state space, an adaptive agent aims to identify an optimal strategy that maximizes average performance $(A)$ and minimizes the average resource consumption $(R)$ by selecting the optimal adaptation state $(S_i)$ for each given input $x$.

We establish the performance and efficiency bounds attainable by any adaptive agent through the concept of an "Oracle Agent", as defined in the subsequent section.

## 2.2 THE ORACLE AGENT

An Oracle Agent is defined as an agent equipped with simultaneous knowledge of both resource consumption and the accuracy (i.e. correctness) of all models for each instance $x$. As a result, it can choose the adaptation state with the lowest resource consumption while still achieving the highest accuracy possible (within the constraints of the adaptation state space) for every classified instance.

In the definition above, the Oracle Agent, like any other adaptive agent, is constrained by the performance and efficiency limits of the corresponding adaptation state space. Consequently, it cannot guarantee correct predictions (or $100\%$ accuracy) for every instance, nor can it achieve greater efficiency than the most efficient state. This contrasts with typical definitions of conceptual Oracles found in literature, but aligns more closely with the capabilities of real-world adaptive agents.

As a conceptual example, consider a 2-state adaptation problem with two backbone classifiers of different sizes:

Consider a larger classifier characterized by $S_L = (R_L, A_L)$ and a smaller model characterized by $S_s = (R_s, A_s)$. In Figure 1, there are only four cases to be considered based on the per-instance accuracy of each classifier. Given that opting for more resources (selecting a larger model) is justified only if it leads to a better relative accuracy, it can be argued that an Oracle Agent would choose the larger model for a specific instance only when the smaller model is inaccurate, i.e., incorrect, while the larger model is accurate, i.e., correct, (as depicted in the IA case in Figure 1).

### 2.2.1 GENERAL FORMULATION

Building upon the insights from this straightforward 2-state scenario, we present the following general definition for an Oracle Agent:

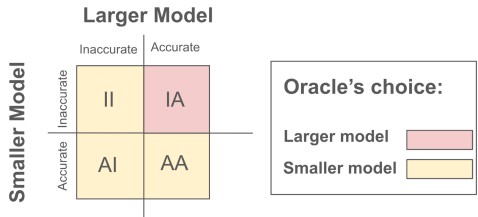

Figure 1: Confusion matrix for a conceptual 2-state classification task. Resource consumption of the Oracle Agent is only a function of $P(IA)$

**Definition 2.1.** Given an adaptation state space $\{S_i\}$, its corresponding $\{R_i\}$ and $\{A_i\}$, and a dataset $X$, the Oracle Agent is defined as an adaptive agent that implements the following strategy:

$$R_{oracle}(x) = \begin{cases} \min_i(R_i) & s.t.\ Y_i(x) = Y_{GT}(x) \\ R_1 & O.W. \end{cases} \tag{4}$$

for all $x \in X$,

Where $Y_i(x)$ is the predicted label from model $i$ on instance $x$, and $Y_{GT}(x)$ is the ground truth label of instance $x$. The expected resource consumption and accuracy achieved by such an Oracle are $R_{oracle}$ and $A_{oracle}$ calculated as:[1]

$$R_{oracle} = R_1(1 - P(e_1) + P(e_N)) + \sum_{i=2}^{N} R_i[P(e_{i-1}) - P(e_i)],$$

$$A_{oracle} = 1 - P(e_N), \tag{5}$$

In which $P(e_i)$ is the probability of event $e_i$ defined as:

$$e_i = \{Y_1 \neq Y_{GT} \cap Y_2 \neq Y_{GT} \cap \cdots Y_{i-1} \neq Y_{GT} \cap Y_i \neq Y_{GT}\},$$

This can be interpreted as the event in which all of the $i$ smallest models fail to classify an instance correctly.

To get a better intuition on the equations above one can use the Bayes rule, and the fact that $A_i = 1 - P(Y_i \neq Y_{GT})$, to write each $P(e_i)$ as:

$$P(e_i) = \begin{cases} \alpha_i(1 - A_i), & i > 1 \\ (1 - A_1) & i = 1 \end{cases} \tag{6}$$

In which $\alpha_i$ is defined for $i > 1$ and can be written as:

$$\alpha_i = P(Y_1 \neq Y_{GT} \cap Y_2 \neq Y_{GT} \cap \cdots Y_{i-1} \neq Y_{GT} | Y_i \neq Y_{GT}).$$

Intuitively, larger $\alpha_i$ values (approaching 1) indicate states where the errors of larger models are inherently challenging to resolve using any of the smaller models. Conversely, a smaller $\alpha_i$ represents the scenario in which an ensemble of smaller models are capable of resolving some or all of the classification errors of a larger model.

Using this definition Equation 5 can be reformulated as:

$$R_{oracle} = R_1 + (R_2 - R_1)(1 - A_1) - \alpha_N(R_N - R_1)(1 - A_N)+$$

$$\sum_{i=3}^{N}[\alpha_{i-1}(R_i - R_{i-1})(1 - A_{i-1})],$$

$$A_{oracle} = 1 - \alpha_N[1 - A_N], \tag{7}$$

The resource consumption and accuracy of an Oracle Agent calculated using this equation can serve as an upper bound on the performance and efficiency achievable by any adaptive agent applied on

---

[1]For a detailed proof of each equation please see the Appendix.

the same adaptation state space. Calculating this upper bound, however, relies on knowledge about the adaptation state space, characterized by $R_i$'s and $A_i$'s of the backbone classifiers, along with the hidden term $\alpha_i$.

For pre-trained off-the-shelf backbone models, $R_i$ and $A_i$ values are typically readily available since they can be calculated separately for each classifier. On the other hand, $\alpha_i$ captures the cross-dependencies among the entire set of backbone models, a detail often not reported for static off-the-shelf models. Obtaining an empirical estimate of $\alpha_i$ while not impossible, necessitates access to both a representative validation set and a comprehensive set of candidate backbone models. This poses a challenge, especially when constructing an adaptive inference pipeline from scratch or when the backbone models undergo frequent retraining to uphold performance amidst real-world data distribution shifts.

Fortunately, for classifiers with similar structure that are trained using the same training set, variations of $\alpha_i$s between states can be relatively small. This allows for calculation of an approximate performance and efficiency bound for Oracle Agents without the need for calculating $\alpha_i$s for the entire adaptation space. This is the motivation for the constant-$\alpha$ formulas investigated in the next section.

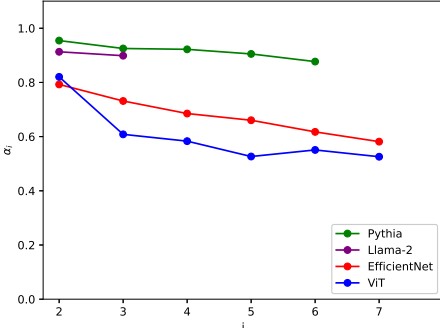

Figure 2: Empirical Measurements of $\alpha_i$ for different tasks and models. $\alpha_i$ remains relatively constant for models with similar architecture.

### 2.2.2 CONSTANT-$\alpha$ APPROXIMATION

As previously discussed, $\alpha_i$ serves as a measure of the probability that a large model making a classification error leads to errors in all of the smaller models.

Intuitively, if the classifiers forming the adaptation state space were statistically independent, one would expect $\alpha_i$ to quickly approach 0 as the number of states increases. This is because as the index $i$ (and subsequently number of states included in calculation of $\alpha_i$) increases, it becomes increasingly more likely that at least one of the smaller models predicts a label correctly by chance.

However, the expected decrease in $\alpha_i$ for larger i values is less pronounced when models forming the state space are not completely independent. In such cases, larger models are anticipated to correctly classify most, if not all, of the samples that are correctly classified by the smaller models. This tendency is commonly observed in backbone classifier families with similar network structures, as demonstrated in Figure 2.

Building on this intuition, one straightforward approach is to assume $\alpha_i$ to be constant and independent of the state index ($\alpha_i = \alpha$). In this scenario, Equation 7 can be modified as:

$$R_{oracle} = R_1 + (R_2 - R_1)(1 - A_1) + \alpha \left[ \sum_{i=3}^{N} [(R_i - R_{i-1})(1 - A_{i-1})] - (R_N - R_1)(1 - A_N) \right],$$

$$A_{oracle} = 1 - \alpha[1 - A_N]. \tag{8}$$

This equation reveals that under the constant-$\alpha$ assumption, the relationship between $R_{oracle}$ and $A_{oracle}$ is a line connecting a very optimistic operating point with $A_{oracle} = 1$ and $R_{oracle} =$

$R_1 + (R_2 - R_1)(1 - A_1)$ (associated with $\alpha = 0$) to a more realistic "conservative" bound with an accuracy of $A_{oracle} = A_N$ corresponding to $\alpha = 1$.

For $\alpha = 1$ we have:

$$R_{oracle} = R_1 + \sum_{i=2}^{N} (R_i - R_1)(A_i - A_{i-1}),$$

$$A_{oracle} = A_N. \tag{9}$$

This equation serves as a conservative estimate for the performance and efficiency gains achievable by any adaptive agent. Moreover, it only requires knowledge about the $R_i$ and $A_i$ values for each state, which are typically readily available for well-known off-the-shelf classifiers.

Table 1: Estimated Adaptation Opportunity Bounds

| Context | Model Family | Baseline State Space | | | Conservative Estimate ($\alpha = 1$) | | | Optimistic Estimate ($\alpha = \alpha_{min}$) | | | | | |
|---|---|---|---|---|---|---|---|---|---|---|---|---|---|
| | | Accuracy Baseline | Efficiency Baseline | | | Efficiency Gain Opportunity | | $\alpha$ Estimate | Accuracy Gain Opportunity | | Efficiency Gain Opportunity | | |
| | | $A_N$ | $R_1$ (GFLOPs) | $R_N$ (GFLOPs) | $R_{oracle}$ (GFLOPs) | $\Delta R$ (GFLOPs) | $R_{ratio}$ | $\alpha_{min}$ | $A_{oracle}$ | $\Delta A$ | $R_{oracle}$ (GFLOPs) | $\Delta R$ (GFLOPs) | $R_{ratio}$ |
| CV (ImageNet) | EfficientNet | 83.95% | 0.39 | 37.75 | 0.60 | 37.15 | **63.43x** | 0.58 | 90.67% | **+6.72%** | 0.54 | 37.21 | **70.26x** |
| | ViT | 88.60% | 4.41 | 1,016.72 | 23.21 | 993.51 | **43.80x** | 0.52 | 94.00% | **+5.66%** | 15.56 | 1,001.16 | **65.33x** |
| | SOTA | 90.88% | 0.04 | 2,586.00 | 21.24 | 2,564.76 | **121.77x** | | | - | | | |
| NLP (HellaSwag) | Pythia | 67.08% | 78.96 | 2,910.00 | 304.42 | 2,605.58 | **9.56x** | 0.88 | 71.13% | **+4.05%** | 286.14 | 2,623.86 | **10.17x** |
| | Llama-2 | 83.79% | 1,670.00 | 17,570.00 | 2,423.36 | 15,146.64 | **7.25x** | 0.90 | 85.43% | **+1.64%** | 2,385.65 | 15,184.35 | **7.36x** |
| | SOTA | 90.96% | 1.90 | 1,352,640.00 | 16,694.40 | 1,335,945.62 | **81.02x** | | | - | | | |

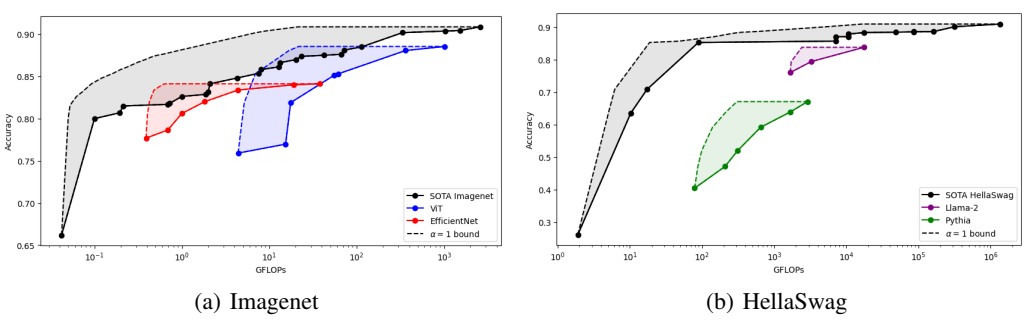

(a) Imagenet

(b) HellaSwag

Figure 3: Operation points achievable by adaptive inference methods under $\alpha = 1$ assumption. The state of the art (SOTA) baseline is used as a proxy for the inherent efficiency versus performance trade-off of each task.

## 3 EXPERIMENTS

In the preceding section, we introduced both exact and approximate bounds aimed at evaluating the achievable efficiency and performance gains of an adaptive agent. This section delves into the practical applications of these bounds on real-world off-the-shelf neural networks.

Our exploration of adaptation spaces focuses on two distinct inference tasks: image classification on ImageNet (Russakovsky et al., 2015) and Natural Language Inference on HellaSwag (Zellers et al., 2019). For each of these tasks, we assessed models tailored for efficiency-critical applications (e.g., image classification at the edge using Efficient-Net) as well as performance-critical applications (e.g., state-of-the-art Llama-2 LLM models deployed on the cloud). Within each state space formed by these models, we present the maximum accuracy achievable by an adaptive agent ($A_{oracle}$) along with the minimum resource consumption required to attain such accuracy ($R_{oracle}$).

The estimated $R_{oracle}$ is then employed to derive two quantitative measures of efficiency gain: $\Delta R = R_N - R_{oracle}$ and $R_{ratio} = R_N / R_{oracle}$, together with $\Delta A = A_{oracle} - A_N$ as a measure of performance gain as detailed in Table 1.

### 3.1 IMAGE CLASSIFICATION BENCHMARK: IMAGENET

ImageNet stands out as one of the most renowned and demanding datasets for image classification, featuring high-resolution images spanning 1000 classes of diverse objects. Off-the-shelf classifiers trained on ImageNet range from compact and efficient models tailored for resource-limited at-the-edge inference such as Efficient-Net (Tan & Le, 2019) to high-performance models typically deployed on the cloud like Vision Transformers (Dosovitskiy et al., 2020).

In Figure 3(a), we present the Performance (Accuracy) versus Resource Consumption (GFLOPs) profiles for two of the prominent pre-trained classifiers on ImageNet, encompassing a broad spectrum of resource requirements and performance capabilities. Additionally, we have calculated the GFLOPs versus accuracy envelope of the state-of-the-art on ImageNet, serving as a proxy for the global adaptation potential of ImageNet (datapoints sourced from the papers-with-code leaderboard (Paperswithcode, 2024)).

Utilizing Equation 9, in conjunction with GFLOPs and accuracy metrics reported in literature for each model, we derived a conservative estimate of the achievable adaptation bounds for each model, as illustrated in Figure 3(a) and summarized in Table 1.

For ImageNet models with a large number of states (e.g., EfficientNet, ViT), even the conservative assumption of $\alpha = 1$ suggests a substantial efficiency improvement potential, in orders of 43-63x. Moreover, the analysis indicates potential for efficiency gains exceeding 121x using the entire state-of-the-art envelope of ImageNet.

### 3.2 NATURAL LANGUAGE INFERENCE BENCHMARK: HELLASWAG

HellaSwag serves as a widely adopted benchmark in the domain of Commonsense Natural Language Inference. Comprising over 10,000 sets of incomplete sentences, each with four potential endings, this dataset tasks language models with selecting the most probable conclusion for a given sentence. The dataset is crafted specifically to necessitate commonsense reasoning based on contextual cues in addition to the words within a sentence.

For this particular task, we chose a large language model typically deployed on cloud infrastructure (Llama-2 (Touvron et al., 2023)), alongside a more compact language model crafted for deployment on resource-limited systems (Pythia (Biderman et al., 2023)). Additionally, we incorporated the GFLOPs vs Accuracy envelope for state-of-the-art models from the HuggingFace LLM leaderboard (Hugginface, 2024; Gao et al., 2023) as a representation of the overarching resource consumption vs accuracy trade-offs associated with HellaSwag. The inference cost of each model (measured in GFLOPs) was computed assuming a batch size of 1 and a maximum sequence length of 128, utilizing the tools provided by Ye (2024) for calculations.

As depicted in Figure 3(b) and detailed in Table 1, state-of-the-art language models show great potential for substantial efficiency improvements through adaptive inference. Specifically, the smaller language model (Pythia) boasts a conservative bound of over 9x in achievable adaptation efficiency gains. Conversely, the larger language model demonstrates the potential for efficiency gains exceeding 7x, a noteworthy accomplishment given the considerable size and scale of such models, resulting in a relative resource consumption reduction of more than 15 TFLOPs. Notably, these accomplishments are surpassed only by the global adaptation potential of the state-of-the-art models on HellaSwag, indicating over 81x potential improvements in efficiency without sacrificing performance.

### 3.3 EMPIRICAL (EXACT) ADAPTATION BOUNDS

The $\alpha = 1$ bounds discussed in the preceding section serve as a conservative estimate on the adaptation potential of various models and datasets. More accurate estimates of the efficiency and performance achievable by an adaptive agent can be obtained by considering the hidden dependencies between models (abstracted by $\alpha_i$) within a specific adaptation state space.

As it was shown in Figure 2, the empirical calculations of $\alpha_i$ for the four state spaces results in relatively constant values, specially since the models have similar structures and training. Using the

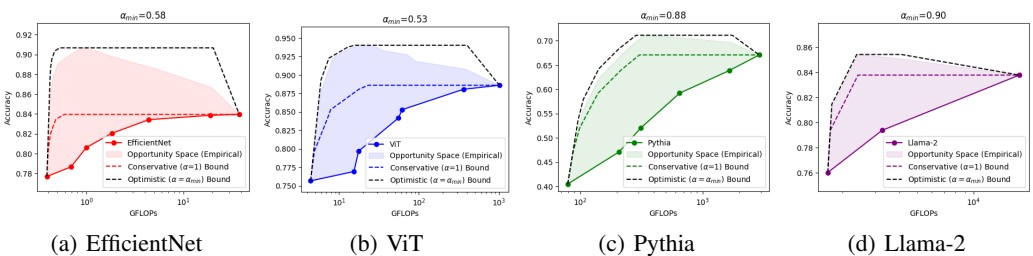

| (a) EfficientNet | (b) ViT | (c) Pythia | (d) Llama-2 |

Figure 4: Proposed $constant - \alpha$ bounds and empirical measurements for an Oracle Agent. The shaded area shows the space of operation points achievable by adaptive inference methods.

empirical measurements of $\alpha_i$ together with Equation 8 one can calculate a constant-$\alpha$ optimistic bound for each model.

As shown in Figure 2, the empirical calculations of $\alpha_i$ across the four state spaces yield relatively constant values, especially given that the models within each adaptation state space have similar structures and training. By combining the empirical measurements of $\alpha_i$ with Equation 8, one can calculate an optimistic efficiency and performance bound for each adaptation state space.

Table 1 showcases the minimum $\alpha$ values measured for each of the four classifier families and their corresponding efficiency and performance gain opportunity bounds. It's crucial to highlight that, unlike the $\alpha = 1$ bounds, which assumed that no accuracy gain are achievable through adaptation, $constant - \alpha$ estimates can be employed to calculate both efficiency and performance gains. For instance, within the EfficientNet family of classifiers, using $\alpha = \alpha_{min}$ to get an optimistic estimate on the performance of an adaptive agent results in an estimated accuracy of $90.67\%$—over $6.72\%$ more accurate comparing to the largest model in the corresponding adaptation state space.

The reported resource consumption values represent the minimum resources required to achieve the performance bounds, indicating that simultaneous improvements in efficiency and performance are attainable for all models based on the $\alpha$ values. For example, the EfficientNet and ViT families (with smaller $\alpha$ values) can achieve accuracy gains of over $5 - 6\%$ while realizing efficiency gains of over 70x and 65x, respectively. On the other hand, tasks related to the HellaSwag dataset exhibit larger $\alpha$ values, resulting in relatively smaller accuracy gains ($4.05\%$ and $1.64\%$ for Pythia and LLama-2, respectively) but still showcasing efficiency gains of over 7-10x for both studied language models.

Figure 4 illustrates the comparison between the approximate bounds and empirical measurements of an ideal Oracle Agent's efficiency and performance across the adaptation space. The visualization highlights that the proposed conservative and optimistic estimates can serve as accurate bounds on the actual operating points achievable by an Oracle Agent (Opportunity Space), particularly for classifier families with larger $\alpha$ values.

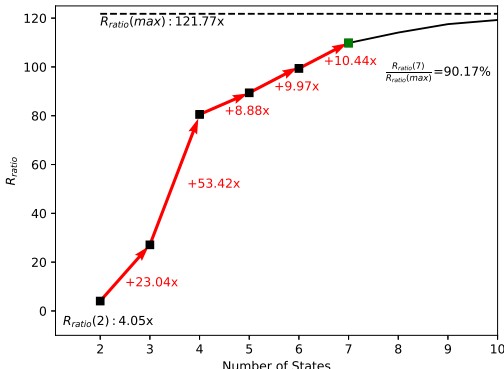

Figure 5: Efficiency gains achievable for discrete state spaces of different sizes. It is possible to achieve 90% of the maximum efficiency gain with only 7 states chosen from Imagenet SOTA.

## 4 ADAPTATION STATE SPACE DESIGN CONSIDERATIONS

In the preceding section, we established the utility of Equation 9 as a measure of adaptation potential for off-the-shelf state spaces. In this section, we aim to provide intuitions and design guidelines for enhancing the adaptation potential of a given adaptation state space.

### 4.1 EFFECT OF STATE SPACE SIZE AND GRANULARITY

Through a simple rearrangement of terms, Equation 9 can be expressed as:

$$R_{oracle} = R_1 + (A_N - A_1)(R_N - R_1) - \sum_{i=2}^{N}(A_i - A_{i-1})(R_N - R_i) \tag{10}$$

The first two terms in this equation represent $R_{oracle}$ calculated using the state dynamic range.[2] However, the third term $\sum_{i=2}^{N}(A_i - A_{i-1})(R_N - R_i)$ is a positive sum that reduces $R_{oracle}$ as each additional state is added, constantly improving the efficiency of the Oracle Agent.

For real-world adaptive systems, expanding the number of states often accompanies increased complexity. Therefore, it is crucial to discern the minimum number of states that result in a sufficient adaptation gain. The following section provides an example demonstrating how Equation 10 can be applied to select a small but effective adaptation space.

#### 4.1.1 OPTIMUM CHOICE OF STATES

Upon revisiting Equation 10, it becomes evident that the efficiency gain resulting from iteratively growing a state space depends solely on the resource consumption of each state ($R_i$) at each step and its accuracy relative to the most similar states existing in the state space from the previous steps ($A_i - A_{i-1}$). Therefore, the utility of each state for all state space sizes can be calculated in linear time.[3]

Figure 5 is evidence that through an optimal design of the state space, a remarkably high adaptation potential can be achieved even within relatively small state spaces. Notably, for ImageNet SOTA, it is possible to realize over a 100x efficiency gain (equivalent to 90% of the efficiency gain of the largest discrete state space) using only 7 states.

#### 4.1.2 OPTIMUM NUMBER OF STATES

The efficiency gain potential figures presented in the preceding sections imply that expanding the state space size exhibits diminishing returns in terms of efficiency gains. Nevertheless, given that Oracle Agents with larger state spaces consistently outperform those with smaller state spaces, considering the concept of an Oracle Agent with access to an infinite number of states ($N \to \infty$ forming a continuous adaptation state space) becomes valuable in theoretically quantifying the achievable efficiency gains for a specific dataset or benchmark.

Let $R_h = \lim_{N \to \infty} R_N$ and $A_h$ be defined as accuracy of the largest state in the continuous adaptation space. The continuous reformulation of Equation 10 can be then derived as:

$$R_{oracle} = R_1 + A_h(R_h - R_1) - \int_{R_1}^{R_h} A(R)\,dR, \tag{11}$$

where $A(R)$ represents the curve depicting the relationship between accuracy and resource consumption in the continuous state space.

As an example, we utilized a continuous piece-wise linear approximation of the SOTA adaptation spaces for ImageNet and HellaSwag to estimate the $\alpha = 1$ bound for continuous adaptation. As presented in Table 2, the gains achievable through continuous adaptation (160.94x and 122.18x for ImageNet and HellaSwag, respectively) significantly exceed the corresponding figures reported in Table 1 for a discrete adaptation space (121.77x and 81.02x respectively).

---

[2]Please refer to the Appendix for design considerations related to the state dynamic range.

[3]For an example of such algorithm please see the Appendix.

Table 2: Conservative bounds for efficiency gain achievable assuming continuous adaptation

| Dataset | $A_{oracle}$ | $R_{oracle}$ (GFLOPs) | $\Delta R$ (GFLOPs) | $R_{ratio}$ |
|---|---|---|---|---|
| **ImageNet** | 90.88% | 16.07 | 2,569.93 | 160.94x |
| **HellaSwag** | 90.96% | 11,070.49 | 1,341,569.52 | 122.18x |

## 4.2 EFFECT OF ADAPTATION COSTS

The Oracle Agent introduced in this work provides an upper bound on efficiency gains achievable for a adaptive inference task independent of the adaptation strategy or system-specific adaptation costs. Examples of such costs in the real-world can include the routing cost in dynamic neural networks or the general cost of switching between different backbone classifiers. However, the proposed framework can easily be modified to include such factors in calculating a more realistic estimate of the efficiency gain potential of specific state spaces.

For example, one simple approach would be to model the adaptation overhead cost as a linear function of the model size and complexity. In other words:

$$\Delta_i = \beta_0 + \beta_1 R_i$$

In which $\Delta_i$ is the adaptation overhead cost for selecting state i, $\beta_0$ is a constant controlling state-independent adaptation overhead costs (e.g. cost of the agent/policy network itself), while $\beta_1$ is a constant controlling state-dependant adaptation overhead costs (e.g. the cost of loading and reloading the neural network weights which is a function of the model size). Adding $\Delta_i$ directly to each $R_i$ the new $R_{oracle}$ can be easily calculated to be:

$$R_{oracle} = (R_1 + \Delta_1)(1 - P(e_1) + P(e_N)) + \sum_{i=2}^{N}((R_i + \Delta_i)(P(e_{i-1}) - P(e_i)))$$

$$= \beta_0 + (1 + \beta_1)R_{oracle}^{-}$$

In which $R_{oracle}^{-}$ is the resource consumption of an oracle with no adaptation cost that can be calculated from Equation 8.

## 5 LIMITATIONS AND FUTURE WORK

The presented work has certain limitations, which will be explored in future research. This includes extending the proposed framework to tasks beyond classification, such as regression, and exploring more advanced models for $\alpha_i$ (e.g., linear instead of constant-$\alpha$) to provide tighter bounds on adaptation potential.

## 6 CONCLUSION

In this work we introduced a novel theoretical framework, quantifying the efficiency and performance gains achievable by adaptive inference methods.

Empirical results demonstrated a substantial efficiency gain opportunity, ranging from 40-70x for models like EfficientNet and ViT (ImageNet), and exceeding 7x (equivalent to a relative computation saving of over 15 TFLOPs) for large language models such as Llama-2. Theoretical estimates for datasets like ImageNet (CV) and HellaSwag (NLP) suggest the potential for achieving over 80-120x efficiency gain using adaptive inference techniques.

Furthermore, we provided insights and design considerations for further enhancing the efficiency gain opportunity by carefully designing the adaptation state space. Empirical results highlight that, for ImageNet, efficiency improvements on the order of 100x can be attained with adaptation space sizes of 7 or less. This paper establishes the theoretical foundation for effective, quantifiable, and systematic design of adaptive inference methods.

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

# A  APPENDIX

## A.1  ASSUMPTIONS

General assumptions (applied to all cases):

1. The adaptation state space $\{S_i\}$ is defined for an ensemble of backbone classifiers.

2. The states $S_i$'s are ranked in an ascending order based on their resource consumption $R_i$.

3. Model accuracies are ranked in an non-decreasing sequence, i.e. $A_i \leq A_j$ if $i < j$.

4. The Oracle Agent has knowledge of both resource consumption ($R_i$) and accuracy ($A_i$) across all models for each instance x, but does not have any knowledge beyond what is provided within the defined adaptation state space.

Conditional assumptions (applied only to certain situations):

1. For Equation 8 and the lower bound Equation 9, assume $\alpha_i = P(Y_1 \neq Y_{GT} \cap Y_2 \neq Y_{GT} \cap \cdots Y_{i-1} \neq Y_{GT} | Y_i \neq Y_{GT})$ is constant across all $i$'s.

2. The constant $\alpha_i$ assumption applies to all results in sections 3 and 4.

## A.2  PROOFS

1. Proof of the general formula equation 5:

   *Proof.* Intuitively, we can estimate the value of $R_{oracle}$ using its expected value, which can be written as:

   $$R_{oracle} = \sum_{i=1}^{N} R_i P(x_i) + R_1 P(x_f),$$
   $$A_{oracle} = 1 - P(x_f), \tag{12}$$

   where $x_1$, $x_i$'s, and $x_f$ are defined as the following events:

   $$x_1 := \{Y_1 = Y_{GT}\},$$
   $$x_i := \{Y_1 \neq Y_{GT} \cap Y_2 \neq Y_{GT} \cap \cdots Y_{i-1} \neq Y_{GT} \cap Y_i = Y_{GT}\},$$
   $$x_f := \{Y_1 \neq Y_{GT} \cap Y_2 \neq Y_{GT} \cap \cdots \cap Y_N \neq Y_{GT}\}.$$

   Then, since $e_i = \{Y_1 \neq Y_{GT} \cap Y_2 \neq Y_{GT} \cap \cdots Y_{i-1} \neq Y_{GT} \cap Y_i \neq Y_{GT}\}$, we see that for $i = 2, 3, \cdots, N - 1$:
   $$P(x_i) = P(e_{i-1}) - P(e_i).$$

   Moreover, it's easy to see that $P(x_1) = 1 - P(e_1)$, and $P(x_f) = P(e_N)$. We can then reformulate the expected value formula in terms of $P(e_i)$'s to get the general formula:

   $$R_{oracle} = \sum_{i=2}^{N} R_i [P(e_{i-1}) - P(e_i)] + R_1 (1 - P(e_1) + P(e_N)),$$
   $$A_{oracle} = 1 - P(e_N). \tag{13}$$

   $\square$

2. Calculations for rewriting the general formula in terms of $\alpha$ (proof of Equation 7):

   *Proof.* Given the general formula:

   $$R_{oracle} = \sum_{i=2}^{N} R_i [P(e_{i-1}) - P(e_i)] + R_1 (1 - P(e_1) + P(e_N)),$$
   $$A_{oracle} = 1 - P(e_N), \tag{14}$$

and Equation 6, we can do the following calculations:

$$R_{oracle} = \sum_{i=2}^{N} R_i[P(e_{i-1}) - P(e_i)] + R_1(1 - P(e_1) + P(e_N))$$

$$= \left(\sum_{i=2}^{N} R_i P(e_{i-1}) - R_i P(e_i)\right) + R_1(1 - P(e_1) + P(e_N))$$

$$= R_2 P(e_1) + \sum_{i=3}^{N} (R_i - R_{i-1})P(e_{i-1}) - R_N P(e_N) + R_1(1 - P(e1) + P(e_N))$$

$$= R_2(1 - A_1) + \sum_{i=3}^{N} (R_i - R_{i-1})\alpha_i(1 - A_i) - R_N \alpha_N(1 - A_N) +$$

$$R_1(1 - (1 - A_1) + \alpha_N(1 - A_N))$$

$$= R_1 - R_1(1 - A_1) + R_1\alpha_N(1 - A_N)) + R_2(1 - A_1) - R_N\alpha_N(1 - A_N) +$$

$$\sum_{i=3}^{N} (R_i - R_{i-1})\alpha_i(1 - A_i)$$

$$= R_1 + (R_2 - R_1)(1 - A_1) +$$

$$\sum_{i=3}^{N} (R_i - R_{i-1})\alpha_i(1 - A_i) - \alpha_N(R_N - R_1)(1 - A_N),$$

which is what we have in Equation 7.

$\square$

3. Proof of the criteria for choosing $R_1$ (Equation 16):

*Proof.* Assume a regular state space $\{S_i\}$ for a set of backbone classifiers. The Oracle Agent's performance is estimated to be:

$$R_{oracle} = R_1 + \sum_{i=2}^{N} (R_i - R_1)(A_i - A_{i-1}),$$

$$A_{oracle} = A_N$$

according to equation 9 with the $\alpha_i = 1$ assumption. Then, assume a special state space $\{S_i'\}$ where all states are identical as in $\{S_i\}$, except that the first state $S_1$ is replaced by $S_1'$, which is a random agent with:

$$R_1' = 0,$$

$$A_1' = \frac{1}{C},$$

where $C$ is the number of classes in the classification task. Then, the Oracle's performance on this special state space is estimated to be:

$$R_{oracle}' = \sum_{i=3}^{N} (R_i)(A_i - A_{i-1}) + R_2(A_2 - \frac{1}{C}),$$

$$A_{oracle}' = A_N.$$

Note here that $A_{oracle} = A'_{oracle}$, so a specific choice of $R_1$ only makes sense if $R_{oracle} < R'_{oracle}$, or if $R_{oracle} - R'_{oracle} < 0$. This then leads to the following computation:

$$R_{oracle} - R'_{oracle}$$

$$= R_1 + \sum_{i=2}^{N}(R_i - R_1)(A_i - A_{i-1}) - \sum_{i=3}^{N}(R_i)(A_i - A_{i-1}) - R_2(A_2 - \frac{1}{C})$$

$$= R_1 + \sum_{i=3}^{N}(R_i - R_1)(A_i - A_{i-1}) + (R_2 - R_1)(A_2 - A_1) -$$

$$\sum_{i=3}^{N}(R_i)(A_i - A_{i-1}) - R_2(A_2 - \frac{1}{C})$$

$$= R_1 + \sum_{i=3}^{N}(R_i)(A_i - A_{i-1}) -$$

$$\sum_{i=3}^{N}(R_1)(A_i - A_{i-1}) - \sum_{i=3}^{N}(R_i)(A_i - A_{i-1}) + (R_2 - R_1)(A_2 - A_1) - R_2(A_2 - \frac{1}{C})$$

$$= R_1 - \sum_{i=3}^{N}(R_1)(A_i - A_{i-1}) + R_2(A_2 - A_1) - R_2(A_2 - \frac{1}{C}) - R_1(A_2 - A_1)$$

$$= R_1 - R_1(A_N - A_2) - R_1(A_2 - A_1) + R_2(A_2 - A_1) - R_2(A_2 - \frac{1}{C})$$

$$= R_1 - R_1(A_N - A_1) - R_2(A_1 - \frac{1}{C})$$

$$= R_1(1 - A_N + A_1) - R_2(A_1 - \frac{1}{C})$$

which gives $R_{oracle} - R'_{oracle} < 0$ if and only if $R_1 < \frac{(A_1 - \frac{1}{C})}{(1 - A_N + A_1)}R_2$, as required in equation 16. $\qquad\square$

## A.3 STATE SELECTION ALGORITHM

As discussed in Section 4, one application of Equation 9 can be used to find the smallest adaptation space with the highest efficiency gain potential given a larger set of possible states. One example of a naive state selection algorithm is shown below.

---

**Algorithm 1** State Selection Algorithm

---
**Input:** Original state space $S$ with size $N$, desired state space size $N_o$ with $N > N_o$,
**Output:** Selected state space $S_o$ with size $N_o$,
Initialize $S_o = \{S_1, S_N\}$.
**for** size $=2$ **to** $N_o$ **do**
    $S_u = S - S_o$
    **for** $s_i$ in $S_u$ **do**
        $j_- = argmax_k[R(s_k)], \qquad s.t. \quad R(s_k) < R(s_i), \ s_k \in S_o$
        $j_+ = argmin_k[R(s_k)], \qquad s.t. \quad R(s_k) > R(s_i), \ s_k \in S_o$
        $dR(s_i) = [R(s_{j_+}) - R(s_i)][A(s_i) - A(s_{j_-})]$
    **end for**
    $s_{new} = argmax_{s_i}[dR(s_i)]$
    $S_o = S_o + \{s_{new}\}$
**end for**

---

## A.4 EFFECT OF STATE DYNAMIC RANGE

Equation 9 reveals a fundamental observation: $R_{oracle}$ is always bounded below by $R_1$, while $A_{oracle}$ is directly tied to $A_N$. Consequently, careful selection of the smallest and largest models in

the adaptation space, defining the *state dynamic range*, is of great importance. Utilizing Equation 15 as a simplified form of Equation 9 for 2 states underscores that adapting with state spaces featuring a broader dynamic range incurs higher efficiency costs but yields more substantial performance benefits.

$$A_{oracle} = A_N,$$
$$R_{oracle} = R_1 + (A_N - A_1)(R_N - R_1). \tag{15}$$

### A.4.1 MAXIMIZING EFFICIENCY: OPTIMUM $S_1$

Given the equation above, the instinctive choice might be to optimize $S_1$ for maximum efficiency (rather than necessarily performance). However, there exists a minimum accuracy threshold for even the smallest state. A practical method to evaluate whether the accuracy of the smallest state justifies its resource consumption is to confirm that the corresponding $R_{oracle}$ is lower than a scenario in which the smallest state provides a random guess on the target class without consuming any resources. Guided by this insight, a design criterion for the resource consumption of the smallest state ($R_1$) can be expressed as :

$$R_1 < \frac{A_1 - \frac{1}{C}}{(1 - A_N + A_1)} R_N \tag{16}$$

In which $C$ is is number of classes in the classification task.

### A.4.2 MAXIMIZING PERFORMANCE: OPTIMUM $S_N$

Unlike $S_1$, the accuracy of the Oracle directly hinges on the accuracy of the most accurate state ($S_N$). Hence, it is logical to craft the largest model in the adaptation space for performance, prioritizing it over efficiency.

As demonstrated in Section 3, attaining higher accuracy for most real-world models often incurs an exponential increase in resource consumption costs. Drawing from Equation 15, the efficiency of the Oracle can exhibit a strong correlation with $R_N$, particularly in state spaces with a larger dynamic range. This implies that the Oracle's resource consumption also grows exponentially relative to its performance. In the subsequent section, we illustrate that increasing the number of intermediate states (increasing state granularity) can alleviate this issue, particularly in scenarios where $R_N$ is exceptionally large.

### A.4.3 STATE SELECTION RESULTS

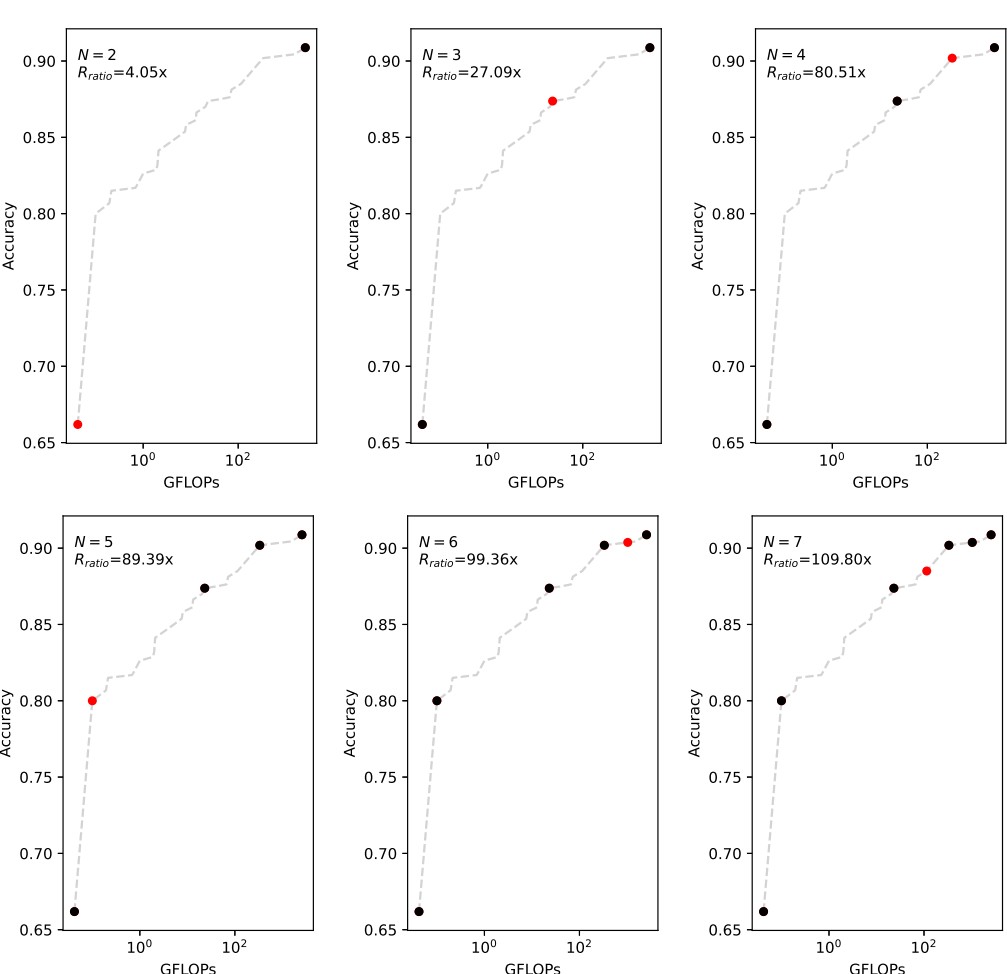

Figure 6: Optimum discrete state spaces of different size and corresponding $R_{ratio}$ for the ImageNet SOTA. The red dot shows the state with the most utility relative to the immediately smaller state space.

