# OpenReview forum: "Adaptive Inference: Theoretical Limits and Opportunities for Efficient AI"
_ICLR.cc/2025/Conference — Submitted to ICLR 2025_

### Official Review · Reviewer_YZbF · 2024-11-02

**Soundness:** 1
**Presentation:** 1
**Contribution:** 2
**Rating:** 1
**Confidence:** 3

**Summary:**

This paper presents a theoretical framework aimed at quantifying the efficiency and performance gains of adaptive inference in neural networks. The authors propose that adaptive inference, where model complexity adjusts based on input instance difficulty, can yield significant efficiency improvements compared to static inference methods. By introducing an adaptation state-space model using Oracle Agents, the authors derive approximate and exact bounds on the achievable efficiency gains of adaptive methods. The study includes empirical demonstrations on models for computer vision and natural language processing, suggesting up to 10-100x efficiency improvements without compromising accuracy. Additionally, the paper offers insights on designing efficient adaptive inference pipelines.

**Strengths:**

1.Theoretical Novelty: The paper makes an original attempt to introduce a theoretical framework for assessing adaptive inference efficiency bounds, potentially contributing to foundational understanding in this area.

2.Potential Impact on Resource Efficiency: If successful, the framework could lead to substantial improvements in resource efficiency, with claimed gains of up to 100x, which is highly relevant for cloud and edge AI deployments.

**Weaknesses:**

1.Writing Clarity Needs Improvement: The paper would benefit from a more polished writing style to improve clarity. For instance, the first sentence of the abstract, "With the commercial deployment of increasingly larger and more complex neural networks at the cloud and the edge in recent years, inference has become too costly in terms of compute workload worldwide," is grammatically awkward and challenging to follow. Enhancing the readability of the text would help convey the contributions more effectively.

2.Unclear Structure and Presentation: The structure is poorly organized, with important theoretical results scattered throughout the text rather than being clearly highlighted as Lemmas or Theorems. This makes it difficult to identify the core theoretical contributions and understand how these results were derived. Additionally, there is no "Related Works" section, and the "Limitations and Future Works" section is brief and lacks organization, which detracts from the paper's overall clarity and coherence.

3.Lack of Clear Experimental Validation: Although the paper theoretically discusses efficiency bounds and includes some empirical comparisons between models (e.g., Llama-2 and Pythia), it is unclear how an Oracle model would achieve the proposed efficiency bounds. The experimental setup lacks clear alignment with the theoretical claims, leaving it ambiguous what specific hypothesis or theoretical results are being validated. This weakens the support for the authors’ claims and raises questions about the practical feasibility of achieving the claimed efficiency improvements.

**Questions:**

My questions are listed in weaknesses part, I hope the authors can give some feedback. Yet, this paper is not easy to follow, which make me hard to get the real point of this work.

---

> ### Author Response · Authors · 2024-11-27
>
> We appreciate the reviewer’s comments and feedback. We would be grateful for more specific and actionable suggestions to improve the work. In particular, further clarification on the reviewer’s concern regarding the alignment between the proposed framework and the empirical results would be highly valuable.
>
> - **Comment 1: Writing Clarity Needs Improvement.**
>
> Thanks for your comment. Following your actionable suggestion we will update the sentence to the following to improve the clarity of the paper:
> “With the rise of larger and more complex neural networks, inference tasks have become increasingly resource-intensive, driving up computational costs globally across cloud and edge deployments.”
>
> - **Comment 2: Unclear Structure and Presentation**
>
> We want to refer the reviewer to the appendix for formal definitions and explanation of each theoretical claim and corresponding assumptions. We also appreciate any specific suggestions on which results or formulations need to be further clarified.
>
> Regarding the need for a related works section, we want to bring to the reviewer’s attention that this is the first theoretical framework in this context to the best of our knowledge. Moreover, although we did not include a separate related works section, we have included several references related to the prior work on adaptive inference in Section1. We would appreciate more specific feedback on what details could be beneficial to include regarding the prior work or the limitations of the presented paper.
>
> - **Comment 3: Lack of Clear Experimental Validation**
>
> We would appreciate the reviewer clarifying why they believe that our claims do not align with the presented results and what specific claim or figure they are referring to.

---

### Official Review · Reviewer_nKmz · 2024-11-02

**Soundness:** 3
**Presentation:** 2
**Contribution:** 2
**Rating:** 5
**Confidence:** 3

**Summary:**

The paper, titled "Adaptive Inference: Theoretical Limits and Opportunities for Efficient AI," addresses the challenge of reducing the computational cost of neural network inference by employing adaptive inference techniques. The authors propose a novel theoretical framework for quantifying the efficiency gains achievable through adaptive inference, contrasting it with traditional static methods such as pruning and quantization. Key contributions include (1) a framework that utilises "Oracle Agents" to establish efficiency and performance bounds, (2) theoretical bounds for efficiency gains, and (3) empirical results showing that adaptive inference can yield significant improvements on tasks such as ImageNet classification and HellaSwag-based natural language inference. The work also provides guidelines for optimizing the "adaptation state space," demonstrating the potential efficiency gains of up to 121x for some models.

**Strengths:**

Below is the list of the strength points identified in this work:

- This paper stands out for its novel approach to quantifying efficiency in adaptive inference. Unlike traditional static techniques, such as pruning and quantization, which primarily focus on model compression, this research introduces a theoretical framework to understand and measure the potential of adaptive inference. The concept of an "Oracle Agent" is particularly interesting, as it provides theoretical bounds for the maximum efficiency achievable through adaptation, making this methodology both original and promising for optimising computational resources.
- The authors back up their theoretical framework with experiments on real-world tasks, such as ImageNet image classification and HellaSwag language inference, showing potential efficiency gains of up to 121x without sacrificing accuracy. These results make the proposed approach more credible and demonstrate that adaptive inference has practical benefits beyond theory.
- The authors provide practical advice for designing adaptive systems, which could be applied across various AI domains, from computer vision to natural language processing.
- By providing both conservative and optimistic estimates of the gains from adaptive inference, the authors make it easier to understand the potential impact of their approach.

**Weaknesses:**

Below is the list of weaknesses that I would like to see refuted or clarified by the authors:

- The framework is based on the idea that the difficulty of an input can be identified quickly, allowing the model to adjust accordingly. However, the paper doesn’t explain how this input complexity could be estimated in a practical setting or what costs this might involve.
- Moreover, the concept of complexity per input sample is central to the adaptive approach but is not clearly defined. How is complexity determined at inference without a priori knowledge?
- The paper often relies on the constant-α approximation, which may oversimplify model dependency in adaptive systems. This assumption may not hold for heterogeneous architectures or diverse datasets.
- The framework implicitly assumes the adaptive gains would generalise across different tasks and data distributions without rigorous testing. Given that models might perform inconsistently across various domains, the generalisation of adaptive efficiency gains seems optimistic.
- The framework is tested on only two datasets (ImageNet for CV and HellaSwag for NLP), which limits the demonstration of its general adaptability across tasks and different dataset dimensions.
- Theoretical derivations and equations, especially around the Oracle Agent, lack intuitive explanations, which might make it difficult for readers to understand the assumptions and implications of key equations.

**Questions:**

Authors are requested to clarify or make changes, as appropriate, based on what is discussed in the ‘Weaknesses’ section.

---

> ### Author Response · Authors · 2024-11-27
>
> We would like to thank the reviewer for their thoughtful feedback and constructive comments. Your insights have been invaluable in helping us refine and strengthen the work. Please find our answers to your questions listed bellow.
>
> **Q1: The paper does not explain how input complexity could be estimated in a practical setting or what costs this might involve**
>
> We want to being to the reviewers attention that this paper's primary contribution lies in establishing theoretical bounds on the efficiency and accuracy gains achievable by adaptive inference methods, rather than proposing a specific method for input complexity estimation.
> Never the less, readers can find several references in Section 1 as examples of practical implementation of adaptive inference techniques. Notable examples include using classification confidence from early exits as an input complexity indicator (as detailed by RA-Net) or leveraging simple policy networks (as demonstrated by AR-Net). These references illustrate feasible approaches for real-time input difficulty assessment.
> Regarding costs related to adaptation policy and ways to incorporate it in the proposed framework please refer to Section 4.2.
>
> **Q2: How is complexity determined at inference without priori knowledge?**
>
> Please refer to the previous question.
>
> **Q3: What if the constant-$\alpha$ assumption does not hold ?**
>
> Thank you for raising this important point. Our framework's general formulation, presented in Equation 7 and Section 2.2.1, does not depend on the constant-$\alpha$ approximation. This assumption is introduced primarily to provide intuitive insights and simplified bounds, which can aid in designing adaptive inference systems, especially when empirical estimation of $\alpha_i$ values is impractical (as illustrated by the state-of-the-art results in Figure 3).
>
> Additionally, our results in Figure 4 and Section 3.3 demonstrate that the proposed bounds remain valid even when there is significant variation in $\alpha_i$​ values across heterogeneous architectures or diverse datasets. This robustness suggests that the constant-$\alpha$ approximation, while simplifying, does not compromise the framework’s applicability or accuracy in realistic scenarios.
>
> **Q4: Concern regarding generalization of the adaptive gains across different task and data distributions**
>
> Thank you for highlighting this concern. In the presented work we assume the test accuracy ($A_i$) and resource consumption ($R_i$) are representative of the actual performance on the adaptive inference test set.
> In cases where discrepancies between training and test distributions are significant, it is standard practice to fine-tune static models using a representative dataset. This fine-tuning not only mitigates accuracy mismatches but also provides reliable estimates of the static accuracies required by our framework. Thus, the proposed methodology remains robust even across different tasks or data domains, provided that adaptive models are appropriately calibrated.
>
> **Q5: Concern regarding testing the framework on only two datasets**
>
> The two datasets and related baselines are only used to provide examples on how the proposed framework can be applied in two well-known target applications. The same process for characterizing adaptation potential as proposed in this paper can be applied to any target task/dataset.
>
> **Q6: The Oracle Agent's derivations and equations lack intuitive explanations**
>
> Thank you for your feedback on the theoretical derivations. We understand the importance of ensuring that the key concepts, particularly those related to the Oracle Agent, are clear and intuitive for readers.
>
> Section 2.2 and Figure 1 are specifically designed to provide an intuitive understanding of the Oracle Agent’s behavior using a simplified 2-state example. Additionally, we introduced the constant-α\alphaα bounds and included detailed experiments in Sections 3.3 and 4, as well as Appendix A.4, to illustrate the practical implications of these equations. These sections aim to bridge the gap between theory and application by offering concrete examples and insights.
>
> We would greatly appreciate more specific feedback regarding which particular equations or derivations you found challenging. This would help us refine our explanations and enhance the clarity of the manuscript in future revisions.

---

> > ### Comment · Reviewer_nKmz · 2024-11-27
> >
> > After reading the comments of the other reviewers and all the authors' responses, I find that although some points are now clearer, the work still requires further study to address the fundamental issues raised. I maintain my original score and my confidence.

---

### Official Review · Reviewer_h41S · 2024-11-03

**Soundness:** 1
**Presentation:** 2
**Contribution:** 3
**Rating:** 3
**Confidence:** 4

**Summary:**

The authors propose a theoretical framework to quantify the efficiency and performance gain opportunity size of adaptive inference methods. In this framework, an Oracle Agent is introduced, which is capable of oracle access to the predictions of an ensemble of classifiers. This agent can then estimate the expected resource consumption and accuracy of the adaptive inference agent using the ensemble. Exact and approximate bounds are provided and evaluated in empirical evidence using computer vision and natural language processing tasks.

**Strengths:**

The authors seem to be the first to tackle the task of adaptive inference from a theoretical perspective, deriving bounds to estimate an achievable resource consumption and accuracy using an Oracle Agent. The paper is well structured, going from the theoretical section to the experimental section.

**Weaknesses:**

**General**
- Since the paper is proposing a *theoretical* framework, it could benefit from more rigorous mathematical definitions and a less convoluted style of expressing, explaining, and reasoning about these. See *Specific* part below.
- There is no related work section.
- Paragraph structures could be improved greatly. There are many paragraphs which consist of a single one or two line sentence.

**Specific**
> However, such efficiency advancements have reached a plateau, necessitating fundamentally new techniques that extend beyond the design space of conventional static neural network optimization methods.

Have they? I'm no expert in that particular field, but I was under the impression that, especially with the rise of *large* language models we saw over the last two years, these fields (compression, pruning, quantization) were thriving. It would be great if the authors would provide relevant sources to these claims.

> Let $A_i$ represent the test accuracy of classifier $i$ represented with $S_i$.

Is this with respect to a specific dataset that the classifier $i$ was trained on (the test set from the same distribution), is this the hypothetical dataset the classifier is being tested/evaluated on when the oracle agent as applied, or what exactly does this accuracy relate to?

> Given the definition of the adaptation state space, an adaptive agent aims to identify an optimal strategy that maximizes average performance ($A$) and minimizes the average resource consumption ($R$) by selecting the optimal adaptation state ($S_i$) for each given input $x$.
>
> [...]
>
> 2.2 The Oracle Agent
>
> An Oracle Agent is defined as an agent equipped with simultaneous knowledge of both resource consumption and the accuracy (i.e. correctness) of all models for each instance $x$. As a result, it can choose the adaptation state with the lowest resource consumption while still achieving the highest accuracy possible (within the constraints of the adaptation state space) for every classified instance.

How is "the optimal adaptation state ($S_i$)" defined? What does it mean to be optimal? Let's say we have $S_1 = (R_1=10, A_1=95%)$ and $S_2 = (R_2=20, A_2=97%)$, which state should be preferred? Doesn't this require a scoring function $f$ that depends on $R_i$ and $A_i$, such that we can actually define an "optimal" state, w.r.t. $f$?


Definition 2.1 states the Oracle Agent strategy ($R_{oracle}(x) = \min_i(R_i) \text{ s.t. } Y_i(x) = Y_{GT}(x)$ and $R_1$ else) which seems to be independent of the classifier accuracy $A_i$? According to Section 2.2, the Oracle Agent strategy is supposed to find "the optimal adaption state" based on the resources $R_i$ **and** the accuracies $A_i$?

Section 2.2.1 goes on and states the *expected* resource consumption $R_{oracle}$ and accuracy $A_{oracle}$ achieved by such an Oracle. This part could greatly benefit from more rigorous mathematical definitions to be able to follow why the authors decide on certain formulations. If we are looking at expectations, why not express the expected resource consumption and accuracy as $\mathbb{E}[R_{oracle}] = \sum_i R_i \cdot P(R_i)$ with $P(R_i)$ being the probability, that the agent chose state $S_i$, which only happens, when all classifiers $< i$ made an error ($E_i := Y_i \neq Y_{GT}$), leading to $P(R_i) = P(E_1, \dots, E_{i-1}, \overline{E_i})$. This is exactly the quantity that the authors define in Appendix A.2 as $P(x_i) = P(e_{i-1}) - P(e_i)$ with $x_i := \{Y_1 \neq Y_{GT} \cap \dots \cap Y_1 = Y_{GT}\}$ (which is an odd naming choice since $x$ is already defined as the datapoint) and $e_i := \{Y_1 \neq Y_{GT} \cap \dots \cap Y_1 \neq Y_{GT}\}$.

Secondly, the authors state the expected accuracy $A_{oracle} = 1 - P(e_N)$ without any derivation or explanation. Since $P(e_N)$ is the probability, that *all* models fail, $1 - P(e_N)$ is the probability, that at least one (or more) model is correct. I cannot follow the reasoning of why this holds, looking at the formulation of the expected value of $A_{oracle}$, we get $\mathbb{E}[A_{oracle}] = \sum_i A_i \cdot P(A_i)$, where $P(A_i) = P(E_1, \dots, E_{i-1}, \overline{E_i})$ which is the probability, that we pick state $S_i$ (same as $P(R_i)$) -- why should this "collapse" to $1 - P(e_i) = 1 - P(E_1, \dots, E_i)$? Appendix A.2 "Proof of Eq. (5)" also does not derive this quantity and simply states the same equation again in Eq. (12).

The authors then go on to rewrite $P(e_i)$ as $\alpha_i (1 - A_i)$ (for $i>1$) with $\alpha_i = P(Y_1 \neq Y_{GT} \cap \dots \cap Y_{i-1} \neq Y_{GT} | Y_i \neq Y_{GT})$ and then rewrite $R_{oracle}$ and $A_{oracle}$ using $\alpha_i$ in Eq. (7). Why are we interested in the quantity $\alpha_i$? Looking at the events, this is the probability, that all classifiers $<i$ fail, given that classifier $i$ failed. The authors state in L202, that larger $\alpha_i$ implies that it is unlikely, that a smaller model can correctly classify what a larger model misclassified and smaller $\alpha_i$ implies that it is likely, that a smaller model can correctly classify what a larger model misclassified -- okay, this is simply how $\alpha_i$ was defined, but why do we need this for the Oracle Agent?

Afterward, the authors claim

> The resource consumption and accuracy of an Oracle Agent calculated using this equation can serve as an upper bound on the performance and efficiency achievable by any adaptive agent

Why exactly is the *expected* resource consumption and accuracy an upper bound on the performance (accuracy?) and efficiency (resource consumption?)?

> On the other hand, $\alpha_i$ captures the cross-dependencies among the entire set of backbone models, a detail often not reported for static off-the-shelf models

Without the authors having defined what they imply by "cross-dependencies among the entire set of backbone models", I assume, that it means that $(Y_j \neq Y_{GT}) \perp\\!\\\!\\!\perp (Y_k \neq Y_{GT})$ (independence) does not hold. I think this is wrong, since $\alpha_i$ is a conditional probability (conditioned on the event $Y_i \neq Y_{GT}$) which then only covers the conditional independence $(Y_j \neq Y_{GT}) \perp\\!\\!\\!\perp (Y_k \neq Y_{GT}) | Y_i \neq Y_{GT}$ with $j < i$ and $k < i$.

> Figure 2: Empirical Measurements of $\alpha_i$ for different tasks and models. $\alpha_i$ remains relatively constant for models with similar architecture.

The figure shows a graph with four plot lines. *All* of them decrease in value $\alpha_i$ with increasing $i$. Saying $\alpha_i$ remains *relatively constant* sounds hand wavy and not scientific at all to me, simply to justify a constant $\alpha$, which is being used in the rest of the body.

> As previously discussed, $\alpha_i$ serves as a measure of the probability that a large model making a classification error leads to errors in all of the smaller models

How does "a large model making a classification error lead to errors in all of the smaller models"? It is simply a probability distribution over events happening (classifiers being correct/wrong), capturing correlation and **not causation**.

I've continued reading the rest of the paper (Section 3 to 6) in detail. Unfortunately, since, as explained above, I'm lost on the reasoning of the choice for most of the definitions in the essential Section 2, I wasn't able to appreciate and follow the results in the later sections.

I hope that I didn't severely misunderstand something which left me confused for the rest of the paper, but as it is written now, the essential Section 2 needs to be greatly improved to clarify most of what I've said above. I'm happy to discuss the details with the authors, and I'm interested to see if other reviewers were able to make sense of Section 2.

**Questions:**

See **Weaknesses**.

---

> ### Author Response · Authors · 2024-11-19
>
> We want to thank the reviewer for their constructive feedback and their detailed questions. We hope that the answers provided below will convince the reviewer to provide more feedback on the other sections of the paper and hopefully appreciate the results and intuitions provided as part of the proposed theoretical framework.

---

> > ### Author Response · Authors · 2024-11-19
> > **Q1: Question about Static model compression gains reaching a plateau**
> >
> > The growing demand for compression methods in recent years serves as a key motivation for the presented work. However, we argue that static methods alone may not suffice to meet the efficiency requirements of large foundational models, as highlighted by the reviewer. Importantly, adaptive inference methods can be **layered on top of static compression techniques**, offering further gains in efficiency and addressing the increasing computational demands of these models.
> >
> > Our argument regarding the efficiency advancements of compression methods reaching a plateau stems from the fact that the efficiency gains reported by the early fundamental papers in the context of computer vision network compression are relevant benchmarks to date. Examples include the ~49x total efficiency gain on VGG16 reported by Han et al in 2015 [1] and 32x quantization-only efficiency gain reported in the XNOR-Net paper in 2016 [2]. While there exist several review papers on recent compression algorithms in the context of computer vision[3] and LLMs[4], to the best of our knowledge none of the recent work in this area have promised the potential for an additional 10-100x improvement in efficiency.
> >
> > [1] Han, Song, Huizi Mao, and William J. Dally. "Deep compression: Compressing deep neural networks with pruning, trained quantization and huffman coding." arXiv preprint arXiv:1510.00149 (2015).
> >
> > [2] Rastegari, Mohammad, et al. "Xnor-net: Imagenet classification using binary convolutional neural networks." European conference on computer vision. Cham: Springer International Publishing, 2016.
> >
> > [3] Li, Zhuo, Hengyi Li, and Lin Meng. "Model compression for deep neural networks: A survey." Computers 12.3 (2023): 60.
> >
> > [4] Chavan, Arnav, et al. "Faster and Lighter LLMs: A Survey on Current Challenges and Way Forward." arXiv preprint arXiv:2402.01799 (2024).

---

> > ### Author Response · Authors · 2024-11-19
> > **Q2: Which dataset does $A_i$ correspond to?**
> >
> > The short answer is that $A_i$ is the (static) accuracy of each Model i on the test-set that adaptive inference method will be ultimately applied to. Our assumption is that the challenges caused by the distribution mismatch between train and test-sets are handled separately through fine-tuning of the backbone models (and the agent) on a representative dataset.

---

> > ### Author Response · Authors · 2024-11-19
> > **Q3: How is optimal adaptation state defined ?**
> >
> > Based on Definition 2.1 it can be seen that the Oracle has knowledge about whether or not each model provides a correct prediction for each input x. This is a stronger assumption than knowledge about $A_i$ of each model.
> >
> > For example, for the case suggested by the reviewer [S1=(10,95),S2=(20,97)], the Oracle follows the procedure explained in Section 2.2 and Figure1, achieving an accuracy better than or equal to $A_2$ (97%) while consuming less power than $R_2$ (20) depending on the $\alpha$ parameter defined in Section 2.2.1.
> >
> > Following the reviewer’s comment we will remove the term “($S_i$)” from the corresponding sentence to avoid confusion. We also welcome any further clarifying questions regarding Section 2.2 and the definition of the Oracle Agent.

---

> > ### Author Response · Authors · 2024-11-19
> > **Q4: Does Oracle's strategy depend on $A_i$?**
> >
> > The Oracle’s strategy does not directly depend on $A_i$. However, the Oracle’s overall accuracy and resource consumption can be written in terms of $R_i$, $A_i$ (and $\alpha_i$).
> > For further clarification please refer to the previous question.

---

> > ### Author Response · Authors · 2024-11-19
> > **Q5: Questions about the notations and the necessity of defining the $\alpha$ parameter**
> >
> > Thanks for your comment about the confusing notation in the appendix. Following your comment we will modify the “$x_i$” notation to avoid confusion.
> >
> > As mentioned by the reviewer and shown in the proof in A.2, both the formulations suggested by the reviewer and the ones used in the paper are equivalent.  We have intentionally decided to use the formulations based on the events $e_i$ instead of the suggested weighted sum formulation to facilitate easier definition of the $\alpha$ parameter later on. The $\alpha$ parameter has several intuitive and practical implications as mentioned in the paper in sections 2.2.1, 2.2.2 and 3.3.
> >
> > To clarify, the Oracle agent’s strategy is not directly relying on $\alpha_i$. However, the expected accuracy and efficiency of the Oracle can be written in terms of $\alpha_i$, $R_i$ and $A_i$.
> >
> > The importance of the current formulation is to separate the effect of two factors affecting $R_{oracle}$ and $A_{oracle}$: $A_i$ and $R_i$ (the static efficiency and accuracy of each individual model i), and $\alpha_i$ (the hidden cross-dependencies between different model’s predictions).
> >
> > As mentioned in the paper in section 2.2.1 (page 5) $R_i$ and $A_i$ can be easily obtained for off-the-shelf models while knowledge about $\alpha_i$ requires analyzing relative behavior of all models simultaneously. Section 3.3 of the paper is dedicated to interpretations of $\alpha$ and related practical implications. As an example we showed that it is impossible to achieve accuracy gains beyond the most accurate classifier for models with $\alpha$ close to 1 while it is easily achievable for models with similar $A_i$ and $R_i$ values but with smaller $\alpha$.

---

> > ### Author Response · Authors · 2024-11-19
> > **Q6: Question about the relationship between $A_{oracle}$ and $P(e_N)$**
> >
> > As mentioned in the paper and pointed out by the reviewer, $P(e_N)$ is the probability that all the models fail in producing a correct prediction. This is also the probability that the second condition in Definition 2.1 holds, in which the Oracle fails to produce a correct prediction for instance x. Therefore the accuracy of the Oracle (the probability that the first condition in Definition 2.1 is met) is $1-P(e_N)$.
> >
> > Please note that the formulation suggested by the reviewer also results in the same conclusion if conditional accuracies are used instead of $A_i$’s. Please let us know if further clarification or a formal proof for this equivalency is needed.

---

> > ### Author Response · Authors · 2024-11-19
> > **Q7: Why are the Oracle's (expected) efficiency and accuracy an upper bound for efficiency and accuracy of any other agent?**
> >
> > We are comparing the expected resource consumption and accuracy of the Oracle to the expected resource consumption and accuracy achievable by other agents. Since the Oracle by definition achieves the highest accuracy (correctness) and efficiency comparing to any other agent for each individual instance, the expected value of its performance (accuracy) and efficiency over the evaluation set are also an upper bounds on efficiency and accuracy of any other agent.

---

> > ### Author Response · Authors · 2024-11-19
> > **Q8: What do the authors imply by "cross-dependencies between models"?**
> >
> > As mentioned in Section 2.2.2:
> >
> > “$\alpha_i$ serves as a measure of the probability that a large model making a classification error leads to errors in all of the smaller models.”
> >
> > Based on the definition of $\alpha_i$ in Section 2.2.1, $\alpha_i$ captures the dependency between two events:
> >
> > 1- The i-th model making a classification error, and
> >
> > 2- All smaller models making a classification error as well.
> >
> > This relationship is what needs to be captured for all models in the adaptation state space and is referred to in this sentence as cross-dependency.
> > We welcome any suggestions on alternative terminology to replace “cross-dependency” that would avoid confusion for the readers.

---

> > ### Author Response · Authors · 2024-11-19
> > **Q9: How do the authors justify the constant-$\alpha$ assumptions based on Figure 2?**
> >
> > The general formulation provided on Equation (7) can be applied to any (nonlinear) $\alpha_i$ curve without the need for any additional assumption. However, as long as the $\alpha_i$ values stay relatively constant and do not quickly fall to 0 (which is the case for the curves shown in Figure 2), the constant-$\alpha$ assumptions can be useful in defining approximate bounds as described in section 2.2.2.
> >
> > In other words, even if the alpha values shown in Figure 2 were not all constant, both the conservative bound (corresponding to $\alpha=1$) and the “optimistic bound” (corresponding to $\alpha=\alpha_{min}$) still hold for each classifier family. This is demonstrated in Figure 4 and Section 3.3 comparing constant-$\alpha$ bounds with empirical measurements for an Oracle agent.
> >
> > Please let us know if any further clarifications are needed on this subject.

---

> > ### Author Response · Authors · 2024-11-19
> > **Q10: Does the $\alpha$ parameter capture correlation or causation between models?**
> >
> > The $\alpha$ parameter does not measure any causal relationship between the models. Following your suggestion we will change the word “lead” to “imply” in the corresponding sentences to avoid confusions.

---

> ### Comment · Reviewer_h41S · 2024-11-25
> **Needs major revision**
>
> I appreciate the author's responses to my main points. While a few things became more clear now, I still think, that the manuscript needs a major revision, especially in Section 2, to address the points I made in my review. I will keep my score and confidence.

---

### Official Review · Reviewer_aDAf · 2024-11-03

**Soundness:** 3
**Presentation:** 3
**Contribution:** 3
**Rating:** 6
**Confidence:** 3

**Summary:**

This paper explores theoretical efficiency and introduces performance bounds for adaptive inference methods, which dynamically adjust network's complexity based on input difficulty. The authors derive these bounds by introducing the concept of an Oracle Agent that optimally selects the least resource-intensive model configuration for each input. To simplify the analysis, the authors further adopt a constant “error correlation", which they justify with empirical evidence.

The paper presents both conservative and optimistic adaptation bounds for various SOTA classifiers on ImageNet and language models on HellaSwag, which highlight the potential accuracy and efficiency improvements achievable through adaptive inference. The authors further discuss optimal design choices for the adaptation state space, and provide insights on the number of states required to maximize efficiency gains.

**Strengths:**

The theoretical framework introduced is novel, and sets the foundation for further theoretical and empirical exploration of adaptive inference methods, which is valuable to the community. The simplifying assumptions adopted in the paper and supported by empirical evidence, ensure that its theoretical findings are practically applicable. Additionally, the conservative and optimistic bounds provided offer insights and guidelines for implementing adaptive inference in real-world models.

**Weaknesses:**

While the theoretical analysis is comprehensive, the paper would benefit from additional experimental validation to further support its claims. Specifically, testing a concrete adaptive inference method, such as AR-Net or early exiting approaches on LLMs, cited in the paper, and verifying whether the observed efficiency-performance trade-off curves satisfy the theoretical bounds, would strengthen the theoretical and empirical foundation. If such experiments are infeasible, providing an explanation would be helpful.

Additionally, in such concrete experiments, it would be beneficial to incorporate adaptation overheads as well, which are currently discussed only theoretically. This would illustrate how they impact the efficiency and performance of adaptive inference methods in more concrete examples.

**Questions:**

- How are the dashed curves (conservative or optimistic bounds) in Figures 3 and 4 produced? Do I understand correctly that they result from applying Equation 9 to different subsets of the state space? Further elaboration on this would enhance the clarity of the paper.
- Could you please clarify more precisely how the shaded regions in Figure 4 are obtained?
- Minor typo: the paragraph in lines 393-397 appears to be repeated.

---

> ### Author Response · Authors · 2024-11-27
>
> We appreciate the reviewer’s detailed and thoughtful feedback.
>
> We hope our responses address your concerns and that the suggested revisions improve the clarity of this work for readers.
>
> **Q1-2: How are the dashed lines and the shaded area in Figure 4 obtained?**
>
> Thanks for the questions. Your assumption is correct. The dashed curves are produced by applying Equation 8 (assuming $\alpha=\alpha_{min}$ and $\alpha=1$ for the optimistic and conservative bounds) to different subsets of the state space. Similarly the shaded region’s boundaries are empirical measurements of the accuracy and efficiency of an oracle Agent (as defined in Definition 2.1) for different subsets of the state space.
>
> Following the reviewers comment we will add the following sentence to Section 3.3 of the paper:
>
> “Figure 4 illustrates the comparison between the approximate bounds and empirical measurements of an ideal Oracle Agent’s efficiency and performance across different state subsets of the adaptation space. Similarly, the dashed curves represent the best efficiency and accuracies obtainable for different subsets of each state space based on Equation 8.”
>
> **Typo on lines 393-397**
>
> Thank you for your comment. We will fix this typo in the final revision of the manuscript.

---

### Official Review · Reviewer_AwYm · 2024-11-04

**Soundness:** 4
**Presentation:** 4
**Contribution:** 3
**Rating:** 6
**Confidence:** 4

**Summary:**

This work provides a theoretical framework for quantifying the adaptive inference opportunity space. The authors introduce a notion of adaptive agents which select between multiple classifiers of varying accuracies and resource costs given data to classify. They introduce the notion of an oracle agent which alway selects the classifier with the lowest resource costs without hampering accuracy. Using the oracle agent, they derive expressions for its resource cost and accuracy as a function of the probability that smaller classifiers overcome errors made by larger models. These expressions form conservative and empirically grounded upper bounds on the resource-accuracy frontier achievable for agents on various datasets.

**Strengths:**

- The goal of the paper—theoretically quantifying adaptive inference opportunities—is both timely and relevant for current industrial applications of ML.
- The authors theoretical framework is explained concisely and written well.
- The bounds appear to match empirical adaptive inference results.
- The theory provides guidance for choosing the number of "states" or classifiers considered by an adaptive agent.

**Weaknesses:**

Theoretical bounds are most useful when they provide a prescription for how to empirically close the gap, but the bounds provided in this work are too generic to provide such insight. Suppose I have an agent with the optimal number of states as defined in Section 4.1.1, but it is not achieving the accuracy-resource trade-off predicted by the theory. How can I relate quantities of interest in my adaptive inference algorithm to terms in Eq. 10?
- See Question 1 and 2 for actionable suggestions

The model provided assumes all costs are part of a single ordered resource value and all objectives are part of a single ordered accuracy value. This oversimplifies the search space and does not allow an agent to balance multiple resource optimizations or consider multiple objectives.
- For example, sparse MoE also facilitates domain adaptation in LLMs (GLaM https://proceedings.mlr.press/v162/du22c/du22c.pdf, Meta-DMoE https://proceedings.neurips.cc/paper_files/paper/2022/file/8bd4f1dbc7a70c6b80ce81b8b4fdc0b2-Paper-Conference.pdf). In these cases, a single accuracy metric seems insufficient to capture the goals of the agent.
- A simpler case is when resources are split across memory and compute. There could be several states in the $(R_i, A_i)$ space that are equivalent on a single resource axis but different when split across two resources axes.
- See Question 3 for actionable suggestions

**Questions:**

1. How could someone map these results to improve an agent? For example, taking a standard sparse MoE agent, how would I identify why my agent fails to achieve better trade-offs as predicted by the theory? Perhaps you can work through an example case-study in the text.
2. If the current theory does not provide prescriptive guidelines for improving an agent, are there changes (e.g. to $\alpha_i$) that could facilitate this? Perhaps expressing $A_i$ and $R_i$ in terms of the specific adaptive inference algorithms' parameters will help. Similar to Question 1, you could either include a case study on this or expand on the discussion in Section 4.2. I recommend referring to a specific adaptive inference technique.
3. Have you considered how this theory would extend to multiple resource or accuracy axes? Are there suggestions you can make for future work (e.g. changes to Eq. 4-7) to facilitate multiple axes?

---

> ### Author Response · Authors · 2024-11-27
>
> We sincerely thank the reviewer for their thoughtful and detailed feedback.
>
> Your insightful comments and constructive suggestions have provided valuable perspectives that will undoubtedly improve the quality of this work.
>
> We hope our responses to your questions highlight additional use cases for the proposed framework and suggest promising directions for future research

---

> > ### Author Response · Authors · 2024-11-27
> > **Q1: How could someone map these results to improve an agent ?**
> >
> > Thank you for the insightful question. Applying the proposed framework to evaluate and enhance specific agents is indeed a promising next step, which we plan to explore further.
> >
> > In general the Oracle’s strategy and corresponding bounds could be used to both evaluate an agent, and guide its training to achieve better adaptation results. We envision two potential ways the framework could assist in this task:
> >
> > - **Choosing between hierarchical and unrestricted Agents:**
> >
> > As mentioned in Section1, there are generally two main approaches to designing adaptive agents, one assumes a sequential or hierarchical classification approach (e.g. early existing methods like RA-Net[1]), while other methods rely on independent (unrestricted) policy networks like the one used in AR-Net[2] or general MoE agents as mentioned by the reviewer.
> >
> > It can be argued that the hierarchical agents are more suitable for state spaces (models) with $\alpha$ closer to 1. This is because they rely on the common assumption that larger (deeper) models are more accurate than smaller(shallower) ones in general. On the other hand the general MoE agents can achieve higher accuracy in case of more diverse agents (corresponding to $\alpha<1$).
> >
> > As an intuitive example, Figure 4c of the paper shows that for the Pythia family trained on HellaSwag, a hierarchical model could in principle achieve similar performance compared to a general unrestricted agent (due to the small gap between the $\alpha=1$ bounds and the optimistic bound). In contrast, for the efficient-Net family trained on Imagenet (Figure 4a), a more unrestricted agent could be a better choice.
> >
> > - **Using the adaptive Oracle to guide the agent during its training phase:**
> >
> > The proposed framework can also be used to compare an agent with the Oracle and potentially train the agent to mimic an Oracle for a specific task.
> > One such approach is analyzing an agent based on its behavior on samples corresponding to events $e_1$ to $e_N$ as defined in Equation 5. Designing the agent’s loss or reward functions based on the distance between the agent’s operating point and the Oracle’s closest estimated operating point (per batch or per event) allows for guiding the agent toward optimal performance by aligning its behavior with theoretical benchmarks.
> >
> > As an example, in a simple 2-state scenario with an agent choosing between two models with different sizes, the agent might be consuming too much resources when trying to analyze very difficult or ambiguous tasks (described by $e_2$ in Equation 5 and the quadrant “II” in Figure 1). In such a scenario one can improve the agent’s efficiency by putting more emphasis on similar samples within the training set either by assigning larger weights to corresponding samples in the loss function or by using non-uniform resampling techniques.
> >
> > While it will be challenging to add a discussion on this topic to the main paper because of space restrictions, we will hint at this research direction as part of the future work and potentially add a relevant example to the Appendix.
> >
> > **References:**
> >
> > [1] Le Yang, Yizeng Han, Xi Chen, Shiji Song, Jifeng Dai, and Gao Huang. Resolution adaptive networks for efficient inference. In Proceedings of the IEEE/CVF conference on computer vision and pattern recognition, pp. 2369–2378, 2020.
> >
> > [2] Yue Meng, Chung-Ching Lin, Rameswar Panda, Prasanna Sattigeri, Leonid Karlinsky, Aude Oliva, Kate Saenko, and Rogerio Feris. Ar-net: Adaptive frame resolution for efficient action recognition. In Computer Vision–ECCV 2020: 16th European Conference, Glasgow, UK, August 23–28, 2020, Proceedings, Part VII 16, pp. 86–104. Springer, 2020.

---

> > ### Author Response · Authors · 2024-11-27
> > **Q2: How can changes to the state space (like $\alpha_i$) facilitate improving an agent ?**
> >
> > In Appendices A.3 and A.4, we provide general guidelines on selecting $R_i​$ and $A_i$​ values to enhance an agent's performance and efficiency. However, the reviewer raises an interesting question regarding the potential benefits of adjusting $\alpha_i$ to further improve an agent's capabilities.
> >
> > Referring to the example provided in Q1, we observe that general unrestricted agents such as the policy network trained in AR-Net can potentially achieve better accuracy compared to the hierarchical agents alternatives. By maximizing the complementary behavior of model’s within an adaptation state space one could ensure a smaller $\alpha$ for a specific task. This could be achieved by including the desired $\alpha$ values as an extra optimization criteria during the training phase of the backbone models.
> >
> > On the other hand, for agents that rely on sequential or hierarchical predictions, like the ones used in RA-Net, it may be advantageous to penalize smaller $\alpha$ values during training. This encourages a more predictable (monotone) state space, which can be crucial for achieving better efficiency in such scenarios.

---

> > ### Author Response · Authors · 2024-11-27
> > **Q3: Can the proposed framework be extended to multiple resource or accuracy axes ?**
> >
> > As pointed out by the reviewer, the proposed framework can not be easily extended to multiple axes of performance. This is specially the case because other measures of performance (e.g. precision, F1-score etc) might not be directly related to the probability of choosing the optimum model ($p(e_i)$ in Equation 6). Additionally, one of the main assumptions in the framework is that the adaptive Oracle is unique. Which is not guaranteed to be the case for general multi-objective adaptive inference scenarios.
> >
> > However, we can envision two approaches to modify the proposed framework for multi-objective adaptive inference scenarios:
> >
> > - The simplest approach to incorporating multiple resource axes into the proposed framework is to combine them into a single measure of resource consumption using a custom scoring function. However, this scoring function must be carefully defined to ensure that the bounds on its expected value remain meaningful. For instance, defining $R_i$​ as a weighted linear combination of latency and power consumption ($R_i = w_1 \times {latency}_i + w_2 \times {power}_i$) allows us to leverage the linear properties of Equations 8 and 9 to calculate analogous bounds for a multi-axis adaptation scenario—assuming the conditions in Equations 2 and 3 of the paper still hold.
> >
> > - A more rigorous approach, however, would involve modifying the Oracle’s strategy to handle more than two optimization axes. The key simplification in the adaptive Oracle’s strategy (as shown in Definition 2.1) rests on the assumption that accuracy always takes priority over efficiency. To extend this to scenarios with multiple optimization objectives, we could introduce intuitive priority rankings among various resource and performance metrics.
> > For example, we might prioritize minimizing latency over reducing average power consumption but rank both of these priorities below maximizing overall accuracy.
> > The main challenge in such a scenario lies in formulating the problem so that the probability of selecting each model ($p(e_i)$) can still be expressed through an intuitive measure like model accuracies. This requires developing a clear mathematical structure that integrates multiple objectives while maintaining interpretability and coherence with the existing framework.

---

> ### Comment · Reviewer_AwYm · 2024-11-29
>
> Thank you for your detailed answer. While the responses to my questions make intuitive sense, the justification seems lacking given this is primarily theoretical paper. As noted by other reviewers, the derivation of the framework is written in minute detail with little space given to actually connecting the theory to real world adaptive agents. Given the page limit, I suggest putting the derivations in the appendix and focusing the main text of discussing the points raised in this thread. A detailed example of a specific adaptive inference strategy would greatly strengthen this work (discussing where the theory reveals missed opportunities and how to address them). Reading the other review discussions, it seems this is inline with a common request.
>
> With regard to multi-objective agents, I agree that relating the different axes to the probability of selecting a given model is not simple. I think it is fine to note that the limitations of the current work in this scenario.
>
> Without a revision containing the suggestion to add an example or a compelling reason it is not needed, I will keep my score as is.

---

### Meta-Review · Area_Chair_DmM3 · 2024-12-13

**Metareview:**

The paper introduces a theoretical framework for adaptive inference algorithms to quantify the efficiency and performance gain opportunity size. To this end, by introducing an oracle agent with access to the predictions of a classifier ensemble, bounds are derived and supported through experimentation on computer vision and natural language tasks. As strengths of the paper, reviewers appreciate the fact that the theoretical findings match empirical evidence, its novelty, and the fact that conservative and optimistic bounds provide insights into practice. The shared critique revolves heavily around the shared reviewers’ belief that the paper requires much more preciseness, clearer writing, and additional empirical support (e.g. quantifying adaptation overhead). In its present form, reviewers indicate that too many assumptions remain unclear, mathematical expressions remain ambiguous, and that writing style is too convoluted. The AC agrees with these points and unfortunately does not see the revision phase to have resolved these concerns. The AC thus recommends to resubmit the  paper in a heavily revised version to a future venue.

**Additional Comments On Reviewer Discussion:**

Reviewers have provided specific, constructive, and very detailed feedback. Among several questions, there were also many concrete suggestions of how to increase clarity, strengthen findings, and improve presentation of the paper. Although some of the questions were addressed in a response, as acknowledged by the reviewers, the majority of reviewers remains unconvinced that the feedback has been adequately addressed. In particular, to tangible revision to the pdf seems to have been made to accommodate the feedback. The AC urges the authors to take the discussion into account in a practical paper re-write

---

### Decision · Program_Chairs · 2025-01-22

Reject